# A Mini-Review Regarding the Clinical Outcomes of In Vitro Fertilization (IVF) Following Pre-Implantation Genetic Testing (PGT)-Next Generation Sequencing (NGS) Approach

**DOI:** 10.3390/diagnostics12081911

**Published:** 2022-08-07

**Authors:** Bogdan Doroftei, Ovidiu-Dumitru Ilie, Nicoleta Anton, Theodora Armeanu, Ciprian Ilea

**Affiliations:** 1Faculty of Medicine, University of Medicine and Pharmacy “Grigore T. Popa”, University Street, no 16, 700115 Iasi, Romania; 2Clinical Hospital of Obstetrics and Gynecology “Cuza Voda”, Cuza Voda Street, no 34, 700038 Iasi, Romania; 3Origyn Fertility Center, Palace Street, no 3C, 700032 Iasi, Romania; 4Department of Biology, Faculty of Biology, “Alexandru Ioan Cuza” University, Carol I Avenue, no 20A, 700505 Iasi, Romania

**Keywords:** next-generation sequencing, pre-implantation genetic testing, aneuploidy, monogenic disorders, structural rearrangements, in vitro fertilization, assisted reproductive technology

## Abstract

Background: PGT-based NGS revolutionized the field of reproductive medicine, becoming an integrated component within current assisted reproductive technology (ART) protocols. Methods: We searched the literature published in the last half a decade in four databases (PubMed/Medline, ISI Web of Knowledge, ScienceDirect, and Scopus) between 2018 and 2022. Results: A total of 1388 articles were filtered, from which 60 met, initially, the eligibility criteria, but only 42 were included (≥100 patients/couples—62,465 patients and 6628 couples in total) in the present mini-review. In total, forty-two (70.0%) reported reproductive outcomes, while eighteen (30.0%) had distinct objectives. Furthermore, *n* = 1, 1.66% of the studies focused on PGT, *n* = 1, 1.66% on pre-implantation genetic testing for monogenic disorders (PGT-M), *n* = 3, 5.0% on pre-implantation genetic testing for structural rearrangements (PGT-SR) and *n* = 55, 91.66% on pre-implantation genetic testing for aneuploidies (PGT-A). Conclusions: PGT using NGS proved to be an excellent companion that folds within the current ascending tendency among couples that require specialty care. We strongly encourage future studies to provide a systematic overview expanded at a larger scale on the role of the PGT-NGS.

## 1. Introduction

Although there is a presumption concerning the intrinsic process of human reproduction, this inherited mechanism is still singular and inefficient. Fortunately, the field of reproductive medicine incorporates an ascending trend, comprising methodologies that brought novel insights [1].

Thus, the current method of choice surrounding assisted reproductive technologies (ARTs) is IVF combined with PGT to minimize the risks of genetically abnormal embryo(s) [2]. Retrospectively, PGT evolved from an experimental procedure performed by Handyside et al. [3] more than three decades ago. The authors clarified the usefulness of PCR amplification to detect repetitive Y sequences in determining the sex of the fetus in families with a history as carriers of X-related malformations [3].

Distinct molecular biology techniques are optimized to respond to this constantly growing trend of couples requiring help and emerge as a countermeasure to the rudimentary protocols [4]. The European Society of Human Reproduction and Embryology (ESHRE) recently updated the terminology. Therefore, pre-implantation genetic diagnosis (PGD) and pre-implantation genetic screening (PGS) became PGT [5].

Presently, PGT may target aneuploidies, monogenic disorders, and structural rearrangements [5], PGT focusing on the biopsy of a single or few cells from fertilized embryos obtained in vitro and tested for potential genetic defects. Even though this procedure is complex and requires resources, next-generation sequencing (NGS) revolutionized the field of reproductive medicine [4].

Specialty societies such as PGD International Society (PGDIS), the American Society for Reproductive Medicine (ASRM), and ESHRE PGT Consortium issued a guideline regarding good practice. ESHRE further published an extent covering technical guidance on PGT organization of biopsied embryos and genetic testing [6,7,8,9].

Therefore, the present manuscript aims to highlight the reproductive outcomes of all studies conducted in the last five years (2018–2022) when combining PGT-NGS in circumstances when equal to or more than 100 patients/couples were enrolled.

## 2. Methodology

The present narrative mini-review follows the standard procedures previously described by Green et al. [10].

### 2.1. Database Search Strategy

The literature database explored for information until inception (April 2022) was PubMed/Medline, ISI Web of Knowledge, ScienceDirect, and Scopus. Several combinations of keywords such as “NGS” in combination with “pre-implantation genetic testing for aneuploidy—PGT-A”, “pre-implantation genetic testing for monogenic disorders—PGT-M”, and “pre-implantation genetic testing for structural rearrangements—PGT-SR” were employed during the databases tracking.

### 2.2. Inclusion Criteria

To emphasize the necessity of performing studies on a larger scale and the clinical relevance, we found it suitable to include ≥100 participants and/or couples. Based on the retrieved results, we created a time series (2018–2022) using Microsoft Excel^®^ 2010 that contains the number of studies per year of publication, number, and database searched. Since PGT-NGS is mostly a human-targeted technique, experiences on experimental models, mice, rats, zebrafish (*Danio rerio*), or other species were not further considered. There were no restrictions concerning diagnosis (private clinics or hospital-based patients), nor in terms of age, sex, religion, and nationality.

### 2.3. Exclusion Criteria

Case report(s)/series, meta-analyses, review(s), standard or systematic, articles written in another language than English, letters to the Editor, conference posters, work protocols, preprints, and computational simulations have not been considered suitable.

### 2.4. Study Selection

Three independent authors (B.D., O.-D.I., and T.A.) screened the titles and abstracts of the retrieved result. We completed the assignment of all the relevant literature based on title, abstract, and full content. Any discrepancy was solved by consent with the remaining two authors (N.A., and C.I.).

### 2.5. Limitations of the Study

We concentrated on a mini-review rather than a quantitative meta-analysis due to the scarcity and heterogeneity of existing evidence.

## 3. Results

In Figure 1 can be found a flowchart of the present study design.

A total of sixty manuscripts in the present manuscript were eligible, from which 75% (*n* = 45) were retrospective studies, 13.33% (*n* = 8) prospective studies, 6.66% (*n* = 4) retrospective observational studies, 3.33% (*n* = 2) randomized controlled trials (RCTs) and 1.66% (*n* = 1) observational study. Only *n* = 42 articles reported reproductive/clinical outcomes, the remaining *n* = 18 being used as informatic support since they differed by objective and results. Due to the plethora of conditions introduced by the authors that define the study design of articles included in this manuscript, we focus on specific data. Therefore, the main parameters of interest are: implantation, pregnancy, clinical pregnancy, ongoing pregnancy, miscarriage, spontaneous pregnancy loss, late pregnancy loss, biochemical pregnancy and ectopic pregnancy. Thus, based on these considerations, we stratified these studies (with or without indications for PGT, respectively, depending on the allocation per groups) according to the demographic information and assignment in numerical order (Table 1 and Table 2).

We also tried to retrieve information related to the sequencer by strictly using article-related data and the supplementary materials section.

(1)Illumina
(1)twelve with MiSeq [11,13,14,16,24,27,29,33,34,38,43,44];(2)three with HiSeq [15,28,32];(3)two with NextSeq [31,41];(4)one with *sequencing-by-synthesis* [45];

(2)Ion Torrent [42]
(1)four with Ion S5 [17,30,40,46];

Following the allocation of the articles that met the eligibility criteria, we extracted several strong points:(1)The body mass index (BMI) and obesity influence the chances of implantation and amplify the risk of miscarriage [40,46], also dependent on the couple’s age [27], cesarean section (C-section) [22], and ovarian reserve [23], rather than correlated with previous unsuccessful pregnancies [36];(2)While age exerts a detrimental effect, it is mitigated through SNP-based PGT-A [39], and the embryos’ morphology possesses a significant threat with greater impact [11,37,47], but contradicted on several occasions [21,29,38]; Embryo morphokinetic [11,34,37] and inner cell mass (ICM) morphology constitute an optimal predictor of sustained implantation [48];(3)Mitochondrial DNA (mtDNA) copy numbers are higher in day 5 blastocysts of older women than day 6 blastocysts, further associated with a low chance of ongoing pregnancies [14,24]; The content of mtDNA is unable to predict the miscarriage risk [12] and additionally refuted when comparing the outcome differences between them [19] despite the cryo-storage [49];(4)Despite the sensitivity of platforms, errors still might occur, and their optimization is mandatory; PGT-NGS significantly improves the clinical outcomes in mosaic embryos [28,41,50], FAST-SeqS being a reliable and scalable PGT-A method [15];(5)Mosaic embryos have poor reproductive potential but retain the ability to result in live births [13,33], further sustaining that TE biopsy did not add detectable adverse effects [42] but as a supplement for the management of recurrent implantation failure (RIF) [45]; However, zona pellucida opening combined with TE biopsy increases the risk of mosaicism [32], TE mosaicism deriving after TE and ICM differentiation [30], while re-biopsy may rescue those with developmental potential [31];(6)Routine endometrial receptivity (ERA) is not supported in patients undergoing first autologous transfer, estradiol (E2) variation before progesterone (P4) initiation without influencing the transfer’s outcome; Estrogen is inversely associated with gestational age [18,51];(7)NGS-based PGT-A ensures good prognosis in patients [35,52] that suffer or are affected by distinct genetic abnormalities such as SMF [43], with structural rearrangements carriers [16,25,26], Turner syndrome [17], and iRPL [44];(8)Public coverage of ART should be strongly encouraged [20].

Although the true potential of the PGT has yet to be elucidated and fully transposed into current protocols, there is still controversy regarding this topic. Fortunately, there are currently 16 RCTs underway, from which six are observational (*n =* 3790 estimated participants) (NCT04734769, NCT04878991, NCT04976920, NCT04732013, NCT04711239, NCT03520933) and 10 are interventional (*n =* 3414 estimated participants) (NCT04414748, NCT04856696, NCT04000152, NCT03900780, NCT05009745, NCT04577560, NCT04485910, NCT04989348, NCT04654741, NCT03530254) (accessed on 4/12/2022) that may offer insight on this matter. Among the observational studies, (*n =* 1) is multicenter, (*n =* 3) reports the experience from a single center, whereas in the remaining (*n =* 2) no center is attributed. Analogous observations are also valid for those interventionals since (*n =* 2) are multicenter, while (*n =* 8) are from a single-center.

### 3.1. Pre-Implantation Genetic Testing for Monogenic Diseases

Unfortunately, between the pre-established timeframe, we could identify only one article aiming to assess the role of aneuploidy in PGT-M in young women. From 364 patients enrolled and subsequently divided into two unequal groups (*n =* 98/*n =* 266), a total of 569 frozen embryo transfer (FET) cycles (*n =* 131/*n =* 438) resulted from 385 oocytes. The aneuploidy screening significantly improved ongoing pregnancy/live birth rates following the first frozen embryo cycles and reduced the associated time for achieving a pregnancy [21].

### 3.2. Pre-Implantation Genetic Testing for Structural Rearrangements

From 1857 blastocysts following 528 cycles in 403 couples enrolled, 216 blastocysts were transferred through FET. There is some controversy regarding the results obtained since there is a significantly higher rate of balanced reciprocal translocation in women than in their counterparts. Additionally, it marks an improvement in transferable blastocysts rate in couples treated with gonadotropin-releasing hormone antagonist (GnRHa) compared with agonist groups [16]. Another study refutes this possible association among the assessed parameters [53]. Similar observations are outlined in a report of unbalanced chromosomal abnormalities, independently of the maternal age and gonadotropin dosage [26]. The risk of unbalanced rearrangement in paracentric and pericentric carriers is in a sex-associated pattern correlated with the ratio of inverted segment size [25].

### 3.3. Pre-Implantation Genetic Testing for Aneuploidy

The prevalence of de-novo segmental aneuploidies is relatively low [54]. Recent evidence brought us closer to resolving the true value of PGT-A [52] despite the necessity of future studies [42]. Four chromosomes (15, 16, 21, and 22) [55] are frequently reported, and identified in abnormalities [56]. SNP-based PGT-A possesses the ability to mitigate the negative effects of maternal age on IVF outcomes [39].

One way to expand this field of research is to optimize the work protocols and increase the accessibility to the general population through distinct national programs. Public coverage of ART procedures could represent a viable option compared with privately funded institutions [20], while different biopsy protocols of the TE may impact the mosaic blastocyst rate [32]. The transfer at the blastocyst stage is preferred, the long-term cryo-storage for more than 36 months remaining a safe alternative that ensures a good prognosis [35,49].

Several parameters modulate the quality of blastocysts by variances of mtDNA content. Interestingly, the associated level did not differ between non- and pregnant women [12], with a mean copy number of 0.0016 ± 0.0012 per genome. Variants of mtDNA can be found in both coding and non-coding regions, affecting the rate of reproductive outcomes, but independently on the maternal age and day of the biopsy [24]. The mtDNA, euploidy rate, and clinical pregnancy rate are superior for D5 compared with D6 blastocysts [14,24], with comparable results in rates between GnRHa and human chorionic gonadotropin (hCG) [57].

There is no relationship between BMI and ploidy, but rather upon semen morphology [27] of the embryo with the mention that the live birth rate can be low to the detriment of high miscarriage rate in obese patients [40,46,58] per BMI classification issued by the World Health Organization (WHO). The BMI and serum P4 had an insignificant impact on the copy number. The level of mtDNA is above the mean by comparison with abnormal chromosomal number phenomenon and following TE biopsy [59]. The clinical outcomes are similar between D5 and D6 blastocysts [19].

Relative telomere length of white blood cells (WBC) could offer insight. There is a correlation between telomere shortening once with aging with the rate of aneuploidy [60] and controlled ovarian stimulation (COS) [61]. This argument is antithetical to the actual rates of euploidy, aneuploidy, mosaicism, or blastocyst formation in men stratified by age [62].

The current body of knowledge does not support the routine of ERA [18] and the E2 supplementation before P4 initiation, being discovered an inverse correlation between E2 priming and pregnancy duration [51].

MALBAC-NGS-PGT-A outweighs MDA-SNP-PGT-A in terms of costs and support [28]. Despite the effectiveness of NGS, aCGH, and SNP array-based PGT in the modern era [41], there is a discrepancy in the applicability and results between the sexes, with often minimum conclusions [43] and errors still might occur [50]. Morphokinetic characteristics of embryos are insufficient [34,63], while cell-free DNA in spent blastocyst culture media might be a reasonable non-invasive approach [64]. Therefore, algorithms such as KIDscore D5 and mathematical models to predict the number of transferable blastocysts begin to be relevant in clinical practice. High platforms with over 95% sensitivity and specificity [15] could ease the effort of clinicians to conduct a genetic consultation [65]. PGT-A not only shortens the time of obtaining a pregnancy and the live birth [66], but oocyte donors ≤25 have similar cycle blastocyst euploidy, formation, and oocyte number retrieved as those between 26 and 30 years [67].

Current PGT-A methods can detect amalgams of euploid and aneuploid cells, which is why several teams of authors had it as an objective to evaluate the benefit of embryo selection. Inner cell mass (ICM) remains the most valuable predictor of sustained implantation [48], alongside pronuclear for [68] ploidy at all ages for euploid embryos of good quality at D5 [47], especially in younger patients. Poor quality does not always imply inadequate competence [29], considering that the STAR offered novel directions that marked the IVF practice worldwide [38].

Data suggest that aneuploid embryos and TE factors of miscarriage even after PGT-A, women still might suffer a pregnancy loss [36] or as direction for RIF management [45]. One can only speculate that patients with iRPL may be prone to clinical miscarriages. This argument highlights two scenarios: either these innovative platforms miss defects within chromosomes, especially at the level of IMC that subsequently leads to miscarriage, or the lack of diagnostic tests for RPL [44]. Moreover, there is a marked reduction of implantation and implicitly of ongoing pregnancies and live birth in women that had a C-section [22].

Despite the scarcity of data in the current literature regarding the impact of ovarian reserve and response on the chromosome status, the odds decline for a biopsied blastocyst to be euploid by 24% in the diminished ovarian reverse (DOR) group compared to non-DOR. The euploidy rates are not affected by the patient’s status and no differences between DOR and non-DOR with regard to living births per transfer were observed [23,69].

Recent studies showed that the transfer of mosaic embryos could give rise to healthy pregnancies, but are risks associated, precisely reduced implantation and high miscarriage rate concomitantly with fetal abnormalities. Therefore, they should not be treated as a priority [33]. Interestingly, low-medium mosaicism in TE arises after the differentiation of TE and ICM [30]. Re-biopsy constitutes the approach that may rescue blastocysts with developmental potential [31] if they exhibit direct or reverse cleavage and are morphologically eligible [11] with approximately half of the live birth rates by comparison with those euploids [13,37]. Recipients of donated oocytes subjected to an embryo transfer at blastocyst stage should be opted instead of cleavage stage embryo transfer as recently demonstrated [70], indicated choice in females with Turner Syndrome in cases with or without mosaicism [17], biopsy not adding additional risks to the neonatal outcomes [71].

## 4. Discussions

We hope this manuscript is a launching pad and may aid different teams of researchers in using it as a support pillar for future large-scale studies and in designing the protocol, possibly, which categories of patients to be enclosed. Although we have tried to cover in both tables as best as possible the reproductive outcomes based on the conditions implied by the authors, several of them had a unique design, while others confirm or refute the previous results of another team. Conclusively, numerous factors may impact reproductive outcomes in a double-edged sword manner, which is why clear inclusion criteria are compulsory to obtain optimal results. Despite the usefulness of predictors to reflect the reproductive potential, risks, and arguably outcome, errors still might occur, relying on sensitivity and specificity of working platforms. Even though national programs should become a priority regardless of the institution, specific interventions have associated risks. In this context, NGS is a groundbreaking research tool with substantial potential in various fields of interest. Thus, the presence of NGS became imperative in all laboratories that conduct ARTs. From our point of view, NGS now stands as the main barrier toward a new stage in our understanding of genetic defects.

Mechanically speaking, NGS is a cluster of novel technologies with a broad spectrum of utility; DNA and RNA sequencing, variants and mutations. It highly surpasses the Sanger sequencing, allowing simultaneous or massively parallel sequencing. Templates equivalent to eight human genomes (25 gigabases) can be sequenced because it does not involve target-specific primers. The foremost advantage by comparison with Sanger is the ability to target multiple sites in a single reaction in contrast to one target per reaction. Platforms designed over the years are 454 pyrosequencing, Illumina, SOLiD, the Polonator, HeliScope Single Molecule Sequencer, Ion Torrent, and PacBio RS. However, NGS involves the fulfillment of five distinct steps; (1) fragmentation, (2) library preparation, (3) massive parallel sequencing, (4) bioinformatic analysis, and (5) interpretation and variant/mutation annotation [72,73,74,75].

Due to its high potential and interest coupled with the recent discoveries, NGS comes tangent with the uprising trend in understanding disease-associated mutations and genetic alterations. The advancements, in particular, target enrichment methods resulted in evidence of variations responsible for dozens of rare genetic diseases; syndromes Schinzel-Giedion, Sensenbrenner, Miller, Kabuki, Fowler, and mutations attributed to hyperphosphatasia mental retardation and neonatal diabetes mellitus [76], but it has certain limitations in studying neurological diseases [77]. The two noninvasive prenatal testing (NIPT) modalities are cell-based/free and assay [78,79]. This technique involves the use of genetic material from maternal blood. Actual options include NGS of cell-free fetal DNA (cffDNA), polymerase-chain-reaction (PCR)-based methods, and microarrays, all attributed to aneuploidy detection and single fetal cell genome analysis. However, certain limitations must be overwhelmed. The possible clinical applications of both qualitative and quantitative NIPT are: qualitative—autosomal dominant and recessive disorders either when the father carries a mutation or compound heterozygosity is present, X-linked and newborn’s preclusion of hemolytic diseases; quantitative—disorders when the mother carriers a mutation or for both parents when they are suspected to be carrying the same mutation [80,81].

Compared with all NGS platforms developed following the fulfillment of the Human Genome Project (HGP) in 2003, only two are operated in clinical laboratories: Ion Torrent and Illumina systems. The remaining were either phased out due to the time/cost ratio or had limitations that could not be confounded. The principle behind Ion Torrent implies exploiting the emulsion PCR utilizing the aboriginal chemistry of deoxyribonucleoside triphosphate (dNTP) that discharge ions during base incorporation by DNA polymerase and a modified silicon chip detecting the pH modification. Illumina implies a concept on the existing Solexa *sequencing-by-synthesis* chemistry and use of the small flow cells, decreased imaging period, and rapid sequencing [73]. Illumina might outweigh the Ion Torrent in sequencing applications, but Ion Torrent could compensate Illumina in filling the gaps in the assembly produced. The main limitation of Illumina except the time required per run is the acquirement of variable fragments in length due to a phase shift of the analyzed sequences and consequently a reduced accuracy in the three ends of the segments. Concerning the Ion platforms, the total data output due to a higher error rate and sequence truncation could be improved [76]. Płoski [82] already provided a comprehensive comparison between Illumina and Ion Torrent platforms. A brief description of the NGS utility is presented in Figure 2.

## 5. Conclusions

In conclusion, this manuscript may be valuable to other groups in designing the working protocol, the allocation of patients, and the previous outcomes in cases of a more extensive study. NGS-PGT may be improved and suitable to respond to other classes of patients that seek specialty care. Fortunately, experiences on obese patients, at an advanced age, based on their history, or suffering from genetic abnormalities start to gain significant interest. We successfully argue that NGS-based PGT revolutionized this field despite the cases where there are no statistically significant differences between the analyzed groups and parameters evaluated. More specifically, genetic testing increases the chances within all parameters of interest established by us, but the possible involvement of stimulation cycle treatment is still studied. The success is sex- and age-dependent, in some situations proving efficient even in >35 years old women. Although biopsy protocols might impact the embryo’s morphology and morphogenetic, coupled with the health status, algorithms and non-invasive protocols to respond to each hypothetical scenario have been created over the years. Cumulatively, PGT using NGS folds on the constantly increasing trend of couples that seek specialty help. In summary, molecular biology techniques should be viewed as integrative components, reflected by the rates of implantation, clinical and ongoing pregnancy and live birth, accompanied by reduced miscarriage.

## Figures and Tables

**Figure 1 diagnostics-12-01911-f001:**
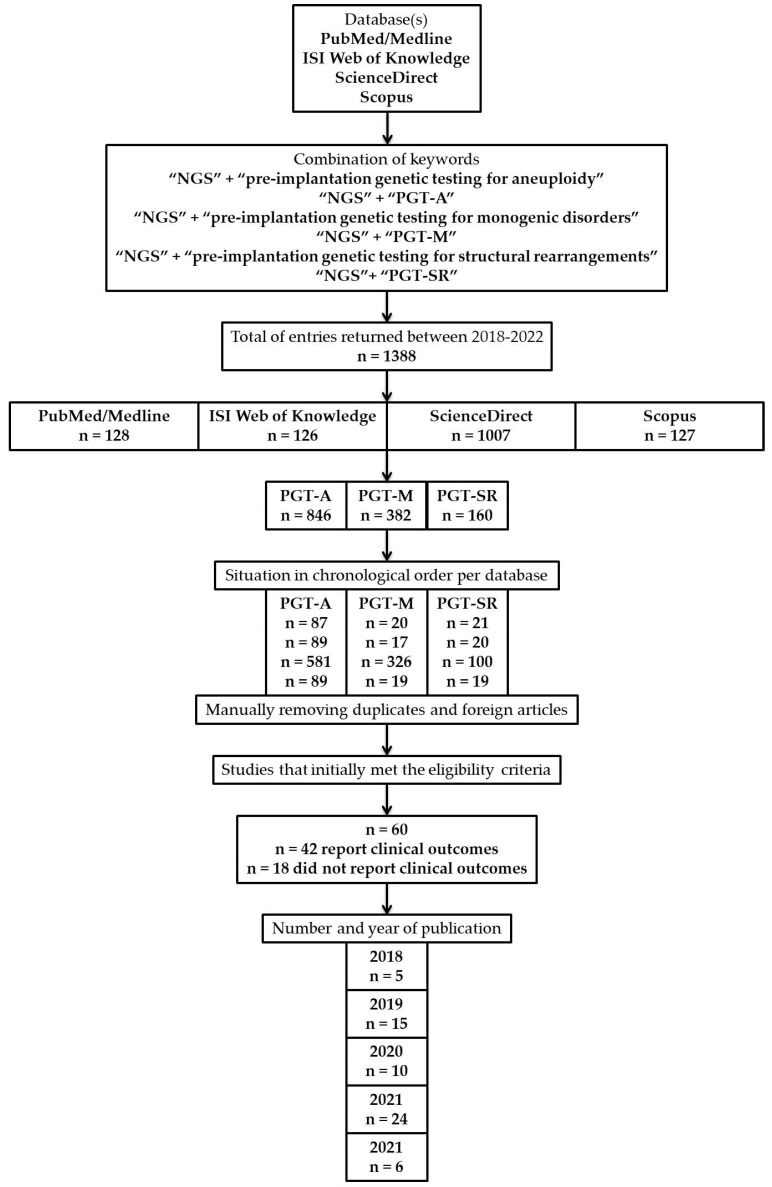
A flowchart of the present study design, strategy, results, and studies that met the eligibility criteria.

**Figure 2 diagnostics-12-01911-f002:**
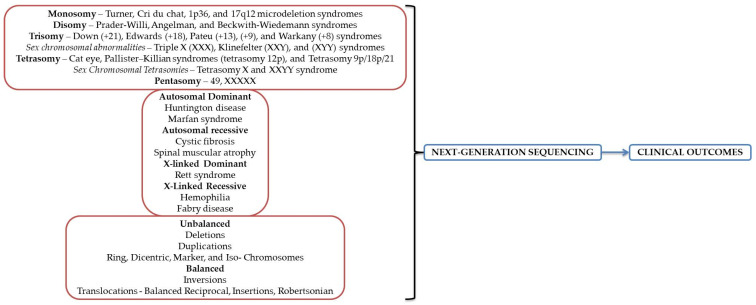
NGS broad utility in genetic disorders diagnosis.

**Table 1 diagnostics-12-01911-t001:** Stratification of studies based on the number of patients/couples grouped without indications for PGT.

No. of Patients or Couples	Reproductive Outcomes									Reference
	Implantation	Pregnancy	Clinical Pregnancy	Ongoing Pregnancy	Miscarriage/ Early Pregnancy Loss	Spontaneous Pregnancy Loss	Late Pregnancy Loss	Biochemical Pregnancy	Ectopic Pregnancy	
**NO ALLOCATION PER GROUPS**										
**270** **patients**				**63.10% (*n* = 53)** **vs.** **46.67% (*n* = 30)**						[11]
**314** **patients**		**66.2% (*n* = 235)**	**52.4% (*n* = 186)**		**5.6% (*n* = 20)**		**2.3% (*n =* 8)**	**7.6% (*n =* 27)**	**0.6% (*n =* 2)**	[12]
**330** **patients**			**60% (*n* = 215) vs.** **40% (*n* = 8)**		**18% (*n =* 65)** **vs.** **40% (*n =* 8)**					[13]
**460** **patients**			**69.7% (*n* = 159) vs.** **57.2 (*n* = 63)**							[14]
**31649 patients**			**62%**							[15]
**100** **couples**			**65.38% (*n =* 34)**	***n =* 8**	**2.94% (*n =* 1)**			**69.23% (*n =* 36)**		[16]
**WITH ALLOCATION PER GROUPS**										
**166** **patients**			**29.0% (*n =* 9)** **vs.** **45.7% (*n =* 64) vs.** **24.0% (*n =* 6)**	**29.0% (*n =* 9)** **vs.** **40.7% (*n =* 57)** **vs.** **16% (*n =* 4)**	**30.9% (*n =* 4)** **vs.** **21.9% (*n =* 16)** **vs.** **42.8% (*n =* 3)**				**0% (*n =* 0)** **vs.** **0% (*n =* 0)** **vs.** **0% (*n =* 0)**	[17]
**228** **patients**			**65.4% (*n =* 53) vs.** **67.4% (*n =* 99)**		**13.2% (*n =* 7)** **vs.** **15.2% (*n =* 15)**			**14.8% (*n =* 9)** **vs.** **15.4% (*n =* 18)**		[18]
**260** **patients**	**67.8% (*n =* 124)** **vs.** **63.6% (*n =* 49)**	**75.4% (*n =* 138) vs.** **70.1% (*n =* 54)**		**57.9% (*n =* 106)** **vs.** **58.4% (*n =* 45)**						[19]
**275** **patients**	***** **31.5%** **vs.** **28.8%** ****** **16.40%** **vs.** **20.50%**									[20]
**364** **patients**	******* **64.29% (*n =* 63)** **vs.** **50.38% (*n =* 134)** ******** **64.12% (*n =* 84)** **vs.** **51.60% (*n =* 226)**				******* **3.17% (*n =* 2)** **vs.** **11.94% (*n =* 16)** ******** **4.76% (*n =* 4)** **vs.** **12.39% (*n =* 28)**			******* **6.12% (*n =* 6)** **vs.** **11.26% (*n =* 17)** ******** **8.70% (*n =* 8)** **vs.** **9.96% (*n =* 25)**		[21]
**525** **patients**	**68.0% (*n =* 221)** **vs.** **55.5% (*n =* 111)**				**13.1% (*n =* 29)** **vs.** **11.7% (*n =* 13)**			**16.9% (*n =* 45)** **vs.** **20.1% (*n =* 28)**		[22]
**1152 patients**					**3.2%** **vs.** **6.8%**			**4.2%** **vs.** **3.9%**	**1.1%** **vs.** **0.4%**	[23]
**142** **couples**	**45.77% (*n =* 65)** **vs.** **29.41% (*n =* 5)**	**59.15% (*n =* 84) vs.** **47.10% (*n =* 8)**		**42.96% (*n =* 61) vs.** **17.65% (*n =* 3)**	**6.15% (*n =* 4)** **vs.** **40.0% (*n =* 2)**			**13.38% (*n =* 19)** **vs.** **17.65% (*n =* 3)**		[24]
**150** **couples**						**3.4% (*n =* 1)** **vs.** **14.7% (*n =* 8)**			**6.9% (*n =* 2)** **vs.** **1.8% (*n =* 1)**	[25]
**180** **couples**				***n =* 9** **vs.** ***n =* 2** **vs.** ***n =* 2**						[26]
**779** **couples**	********* **67.9% (*n =* 106)** **vs.** **69.6% (*n =* 117)** **vs.** **75.6% (*n =* 68)** ********** **86.8% (*n =* 33)** **vs.** **78.4% (*n =* 29)** **vs.** **46.4% (*n =* 13)**				********* **51.2% (*n =* 66)** **vs.** **47.4% (*n =* 65)** **vs.** **62.2% (*n =* 51)** ********** **44.7% (*n =* 17)** **vs.** **64.7% (*n =* 22)** **vs.** **69.2% (*n =* 18)**					[27]
**1418 couples**			**50.5% (*n =* 341) vs.** **41.7% (*n =* 228)**		**15.5% (*n =* 54)** **vs.** **22.8% (*n =* 52)**					[28]

*—Chromosome translocation patients, **—aneuploidy patients, ***—FET with or without aneuploidy screening in the first ET attempt, ****—FET with or without aneuploidy screening in all transfer attempts, *****—<40 y, ******—≥40 y.

**Table 2 diagnostics-12-01911-t002:** Stratification of studies based on the number of patients/couples with indications for PGT.

No. of Patients or Couples	Reproductive Outcomes									Reference
	Implantation	Pregnancy	Clinical Pregnancy	Ongoing Pregnancy	Miscarriage/ Early Pregnancy Loss	Spontaneous Pregnancy Loss	Late Pregnancy Loss	Biochemical Pregnancy	Ectopic Pregnancy	
**NO ALLOCATION PER GROUPS**										
**296 patients**	**85.7% (*n =* 12)** **vs.** **84.0% (*n =* 84)** **vs.** **80.0% (*n =* 44)** **vs.** **80.0% (*n =* 8)**				**7.1% (*n =* 1)** **vs.** **6% (*n =* 6)** **vs.** **5.4% (*n =* 3)** **vs.** **10% (*n =* 1)**					[29]
**783 patients**					**12.0% (*n =* 29)** **vs.** **11.0% (*n =* 15)** **vs.** **12.7% (*n =* 8)**			**10.7% (*n =* 29)** **vs.** **12.3% (*n =* 19)** **vs.** **13.7% (*n =* 10)**		[30]
**1531 patients**			**44.4% (*n =* 8)**	**38.9% (*n =* 7)**						[31]
**WITH ALLOCATION PER GROUPS**										
**206 patients**			**64.71% (*n =* 22) vs.** **65.71% (*n =* 23)**		***n =* 1** **vs.** ***n =* 3**					[32]
**108 patients**	**51.8% (*n =* 43)** **vs.** **52% (*n =* 13)**		**47% (*n =* 39)** **vs.** **52% (*n =* 13)**	**47% (*n =* 39)** **vs.** **36% (*n =* 9)**	**5.1% (*n =* 2)** **vs.** **30.7% (*n =* 4)**					[33]
**108 patients**	**79.4% (*n =* 50)** **vs.** **66.7% (*n =* 16)** **vs.** **25.0% (*n =* 5)**		**76.2% (*n =* 48) vs.** **62.5% (*n =* 15) vs.** **25.0% (*n =* 5)**	**68.3% (*n =* 43) vs.** **62.5% (*n =* 15) vs.** **10.0% (*n =* 2)**						[34]
**155 patients**	**80% (*n =* 68)**		**66% (*n =* 56)**							[35]
**283 patients**				**52.3% (*n =* 148)**						[36]
**554 patients**			**63.9% (*n =* 186)**		**19.3% (*n =* 36)**					[37]
**661 patients**				**50.0% (*n =* 137) vs.** **45.7% (*n =* 143)**	**9.9% (*n =* 27)** **vs.** **9.6% (*n =* 30)**			**10.6% (*n =* 29)** **vs.** **8.3% (*n =* 26)**		[38]
**974 patients**	**69.9%** **vs.** **64.9%**		***n =* 472** **vs.** ***n =* 94**		***n =* 21** **vs.** ***n =* 6**					[39]
**1051 patients**					**14.5% (*n =* 100)**			***n =* 68**	***n =* 11**	[40]
**1513 patients**	**51.34% (*n =* 306) vs.** **49.56% (*n =* 227)**					**10.07% (*n =* 60) vs.** **6.33% (*n =* 29)**		**9.56% (*n =* 57)** **vs.** **10.48% (*n =* 48)**		[41]
**648 couples**	**64.7% (*n =* 202)** **vs.** **0% (*n =* 0)** **vs.** **68.8% (*n =* 11)** **vs.** **30.8% (*n =* 12)**		**73.1%** **vs.** **23.5%**		**7.4% (*n =* 23)** **vs.** **23.5% (*n =* 24** **vs.** **12.5% (*n =* 2)** **vs.** **5.1% (*n =* 2)**			**9% (*n =* 28)** **vs.** **16.7% (*n =* 17)** **vs.** **12.5% (*n =* 2)** **vs. 12.8% (*n =* 5)**	**1.0% (*n =* 3) vs.** **0% (*n =* 0)** **vs.** **0% (*n =* 0)** **vs.** **0% (*n =* 0)**	[42]

## Data Availability

The datasets used and analyzed during the current study are available from the corresponding author upon reasonable request.

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
