# Peer review of "A Mini-Review Regarding the Clinical Outcomes of In Vitro Fertilization (IVF) Following Pre-Implantation Genetic Testing (PGT)-Next Generation Sequencing (NGS) Approach"

_diagnostics, 2022, doi:10.3390/diagnostics12081911_

Round 1

Reviewer 1 Report

The authors present a review about the clinical outcomes following PGT-NGS approach. Although it is a well written manuscript, several corrections should be made to achieve better comprehension. First of all, the title of the manuscript should be simplified as "A Review Regarding the Clinical Outcomes Following PGT-NGS Approach". Second, the whole manuscript should be edited by a native English speaker who is also familiar with medical terms and scientific paper writing. Third, I recommend that the section entitled as "Future perspectives and directions of research" should be omitted from the manuscript as it is irrelevant to the content and scope of the manuscript as it is a retrospective narrative of the experience about PGT-NGS approach. 

Author Response

Dear Reviewer #1,

We would like to thank you very much for the positive feedback, interest, and time spent reviewing our manuscript. Per your instructions, we made the respective changes that can be found below:

Comments from the Reviewer: The authors present a review about the clinical outcomes following PGT-NGS approach. Although it is a well written manuscript, several corrections should be made to achieve better comprehension.

Response: Dear Reviewer, thank you very much.

Comments from the Reviewer: First of all, the title of the manuscript should be simplified as "A Review Regarding the Clinical Outcomes Following PGT-NGS Approach".

Response: Dear Reviewer, since we received multiple suggestions to change the title of the manuscript, we combined all indications and obtained the following title: A Mini-Review Regarding the Clinical Outcomes of In Vitro Fertilization (IVF) Following Pre-Implantation Genetic Testing (PGT)-Next Generation Sequencing (NGS) Approach”.

Comments from the Reviewer: Second, the whole manuscript should be edited by a native English speaker who is also familiar with medical terms and scientific paper writing.

Response: Dear Reviewer, the entire manuscript was edited, renewed with medical terms, and in a scientific writing style.

Comments from the Reviewer:  Third, I recommend that the section entitled as "Future perspectives and directions of research" should be omitted from the manuscript as it is irrelevant to the content and scope of the manuscript as it is a retrospective narrative of the experience about PGT-NGS approach.

Response: Dear Reviewer, per your instructions, that section was removed and will not appear in the final version of the manuscript.

Kind regards and all the best,

Ovidiu-Dumitru Ilie

Reviewer 2 Report

The manuscript "A Retrospective Narrative Mini-Review Regarding the Clinical Outcomes Following PGT-NGS Approach" is an interesting mini-review of the clinical outcomes of all studies conducted in the last five years (2018-2022) when combining PGT-NGS. The field of the review has great interest to a general audience, but it looks confusing in some points of the text, so they need to be clarified.

Table 1 looks confusing, there are too much information and data, which need to be summarized and clarified. The conclusion is not univocal: the authors conclude that "compared with other molecular markers in RIF, emphasis should be placed on the ART-vaginal microflora relationship"; it looks not correspondent with the aim of the work. The authors declare that "Unfortunately, there is a lack of heterogeneity between the studies": is it correct as concept? please re-modulate this phrase.

In the abstract PGT-M, PGT-SR and PGT-A should be spelt out as used for the first time.

The title is not explicative: I suggest modifying it into "A Retrospective Narrative Mini-Review Regarding the Clinical Outcomes of In Vitro Fertilization (IVF) Following PGT-NGS Approach"

The authors performed an accurate update of the literature but they need to specify if it was done conforming to PRISMA guidelines (PRISMA (prisma-statement.org). A figure with the flow chart of the choice of the papers included in the review is mandatory. 

The authors have not adequately highlighted the strengths and limitations of their study. I suggest better specifying these points in the discussion and the conclusion of this work.

What are the actual clinical implications of this study? it is important to report the results obtained by the authors in the context of clinical practice and to adequately highlight what contribution this study adds to the literature already existing on the topic and to future study perspectives.

Author Response

Dear Reviewer #2,

We would like to thank you very much for the positive feedback, interest, and time spent reviewing our manuscript. Per your instructions, we made the respective changes that can be found below.

Comments from the Reviewer: The manuscript "A Retrospective Narrative Mini-Review Regarding the Clinical Outcomes Following PGT-NGS Approach" is an interesting mini-review of the clinical outcomes of all studies conducted in the last five years (2018-2022) when combining PGT-NGS. The field of the review has great interest to a general audience, but it looks confusing in some points of the text, so they need to be clarified.

Response: Dear Reviewer, thank you very much.

Comments from the Reviewer: Table 1 looks confusing, there are too much information and data, which need to be summarized and clarified.

Response: Dear Reviewer, Table 1 suffered significant changes. We divided Table 1 into two small tables but further removed unnecessary data and information to create a better view. We focused mainly on nine parameters of interest: implantation, pregnancy, clinical pregnancy, ongoing pregnancy, miscarriage/early pregnancy loss, spontaneous pregnancy loss, late pregnancy loss, biochemical pregnancy, and ectopic pregnancy. We further maintain a delimitative pointer between those that had indications for PGT or not, without deviating from the main topic, respectively, allocation per groups or not.

Comments from the Reviewer: The conclusion is not univocal: the authors conclude that "compared with other molecular markers in RIF, emphasis should be placed on the ART-vaginal microflora relationship"; it looks not correspondent with the aim of the work.

Response: Dear Reviewer, per the instructions received from the second Reviewer also, we removed the entire section dedicated to vaginal microflora and all additional references.

Comments from the Reviewer: The authors declare that "Unfortunately, there is a lack of heterogeneity between the studies": is it correct as concept? please re-modulate this phrase.

Response: Considering the major changes in the text, that sentence has no longer been considered valuable and, therefore, was removed.

Comments from the Reviewer: In the abstract PGT-M, PGT-SR and PGT-A should be spelt out as used for the first time.

Response: Dear Reviewer, all abbreviations are defined based on how they first appear in the text.

Comments from the Reviewer: The title is not explicative: I suggest modifying it into "A Retrospective Narrative Mini-Review Regarding the Clinical Outcomes of In Vitro Fertilization (IVF) Following PGT-NGS Approach".

Response: Dear Reviewer, since we received multiple suggestions to change the title of the manuscript, we combined all indications and obtained the following title: A Mini-Review Regarding the Clinical Outcomes of In Vitro Fertilization (IVF) Following Pre-Implantation Genetic Testing (PGT)-Next Generation Sequencing (NGS) Approach”.

Comments from the Reviewer: The authors performed an accurate update of the literature but they need to specify if it was done conforming to PRISMA guidelines (PRISMA (prisma-statement.org). A figure with the flow chart of the choice of the papers included in the review is mandatory. 

Response: Dear Reviewer, although we did not conduct a meta-analysis to be obliged to respect PRISMA guidelines, we did however add a flowchart in the revised version of the manuscript.

Comments from the Reviewer: The authors have not adequately highlighted the strengths and limitations of their study. I suggest better specifying these points in the discussion and the conclusion of this work. What are the actual clinical implications of this study? it is important to report the results obtained by the authors in the context of clinical practice and to adequately highlight what contribution this study adds to the literature already existing on the topic and to future study perspectives.

Responses: Dear Reviewer, we added several sentences in the revised version as follows:

Discussion - We hope this manuscript is a launching pad and may aid different teams of researchers in using it as a support pillar for future large-scale studies and in designing the protocol, possibly, which categories of patients to be enclosed. Although we have tried to cover in both tables as best as possible the reproductive outcomes based on the conditions implied by the authors, several of them had a unique design, while others confirm or refute the previous results of another team. Conclusively, numerous factors may impact reproductive outcomes in a double-edged sword manner, which is why clear inclusion criteria are compulsory to obtain optimal results. Despite the usefulness of predictors to reflect the reproductive potential, risks, and arguably outcome, errors still might occur, relying on the sensitivity and specificity of working platforms. Even though national programs should become a priority regardless of the institution, specific interventions have associated risks.  In this context, NGS is a groundbreaking research tool with substantial potential in various fields of interest. Thus, the presence of NGS became imperative in all laboratories that conduct ARTs. From our point of view, NGS now stands as the main barrier toward a new stage in our understanding of genetic defects.

 Conclusions - In conclusion, this manuscript may be valuable to other groups in designing the working protocol, the allocation of patients, and the previous outcomes in cases of a more extensive study. NGS-PGT may be improved and suitable to respond to other classes of patients that seek specialty care. Fortunately, experiences on obese patients, at an advanced age, based on their history, or suffering from genetic abnormalities start to gain significant interest.  We successfully argue that NGS-based PGT revolutionized this field despite the cases where there are no statistically significant differences between the analyzed groups and parameters evaluated. More specifically, genetic testing increases the chances within all parameters of interest established by us, but the possible involvement of stimulation cycle treatment is still studied. The success is sex- and age-dependent, in some situations proving efficient even in >35 years old women. Although biopsy protocols might impact the embryo’s morphology and morphogenetic, coupled with the health status, algorithms and non-invasive protocols to respond to each hypothetical scenario have been created over the years. Cumulatively, PGT using NGS folds on the constantly increasing trend of couples that seek specialty help. In summary, molecular biology techniques should be viewed as integrative components, reflected by the rates of implantation, clinical and ongoing pregnancy, and live birth, accompanied by reduced miscarriage.

Kind regards and all the best,

Ovidiu-Dumitru Ilie

Round 2

Reviewer 2 Report

The authors have done a good job, respecting all the suggestions indicated by the authors. Now the manuscript is suitable for publication

This manuscript is a resubmission of an earlier submission. The following is a list of the peer review reports and author responses from that submission.

Round 1

Reviewer 1 Report

Comments:

  1. Authors does not explain Next-Generation Sequencing and its significance in the diagnosis of genetic disease.
  2. Line number 37 and 38 is not too clear.
  3. In the section “Pre-implantation Genetic Testing for Aneuploidy” author add figure to represent the different genetic disease for better understanding and to make it interesting.
  4. In table 1, in few section authors does not mention the value of “n=?”.
  5. Line no. 102 is not clear.
  6. Authors mention name of different types of NGS, it should be explain on the basis of their difference, applications and limitations in diagnosis (mainly genetic diseses).

Author Response

Dear Reviewer #1,

Please consult the attached Response Letter regarding your comments.

Kind regards and all the best,

Ovidiu-Dumitru Ilie

Reviewer 2 Report

The title of this minireview is: A Retrospective Narrative Mini-Review Regarding the Clinical Outcomes Following PGT-NGS Approach. Is very hard in the conclusion to understand what are the results of the review. Moreover discussion is completely missing. A sort of comments are in the results section but all the paper is quite difficult to understad, despite sthe selection of the articles is well conducted. 

Author Response

Dear Reviewer #2,

Please consult the attached Response Letter regarding your comments.

Kind regards and all the best,

Ovidiu-Dumitru Ilie

Reviewer 3 Report

Pre-implantation genetic testing (PGT)-based next-generation sequencing (NGS) is a comment assisted reproductive technology to test the potential genetic defects. This review enrolled 42 studies combining PGT-NGS in the past five years to present the clinical outcomes.

Specific points:

1. Introduction:

The introduction was appropriate.

2. Material and Methods:

a. The inclusion criteria should be more detailed.

b. In the ‘study selection’, authors should indicate the specific parameters of clinical outcomes which they collected.

3. Results:

The search strategies were comprehensive and the included researches were enlarged size. However, some details should be improved.

a. In the first paragraph of Results, some basic information about the included studies were also mentioned in the Figure 1, such as the published year. Authors should avoid repetitive content.

b. In Table 1, authors mentioned the clinical outcomes of each included researches. However, authors did not explain the object for comparison in each study.

c. Table 1 should be more concise.

4. Discussion

a. It’s a little bit short for the discussion. The significance of this review for clinicians should be emphasized.

Author Response

Dear Reviewer #3,

We would like to thank you very much for the positive feedback, interest, and time spent reviewing our manuscript. Per your instructions, we made the respective changes that can be found below:

Comments from the Reviewer:

Pre-implantation genetic testing (PGT)-based next-generation sequencing (NGS) is a comment assisted reproductive technology to test the potential genetic defects. This review enrolled 42 studies combining PGT-NGS in the past five years to present the clinical outcomes.

Specific points:

  1. Introduction:

The introduction was appropriate.

Response:

Dear Reviewer, we thank you for your feedback.

Comments from the Reviewer:

  1. Material and Methods:
  2. The inclusion criteria should be more detailed.
  3. In the ‘study selection’, authors should indicate the specific parameters of clinical outcomes which they collected.

Response:

a)

Dear Reviewer, the following paragraph has been added into the manuscript:

To accentuate the clinical relevance, we found it suitable to include ≥100 participants and-or couples. Based on the retrieved results, we created a timeline (2018-2022) using Microsoft Excel® 2010 that contains the number of studies per database searched. Since PGT-NGS is mostly a human-targeted technique, experiences on experimental models, mice, rats, zebrafish (Danio rerio), or other species were excluded. There were no restrictions concerning diagnosis (private clinics or hospital-based patients), nor in terms of age, sex, religion, and nationality.”

b)

Dear Reviewer, the following paragraph has been added into the manuscript:

The four main parameters of interest were: implantation, clinical and ongoing pregnancy, and miscarriage rates. Based on the study design, where there were other variables that might impact the clinical outcomes derived from the parameters listed above were also included (e.g. rates per embryo transfer, cycle, or patient).”

Comments from the Reviewer:

  1. Results:

The search strategies were comprehensive and the included researches were enlarged size. However, some details should be improved.

  1. In the first paragraph of Results, some basic information about the included studies were also mentioned in the Figure 1, such as the published year. Authors should avoid repetitive content.
  2. In Table 1, authors mentioned the clinical outcomes of each included researches. However, authors did not explain the object for comparison in each study.
  3. Table 1 should be more concise.

Response:

  1. a) Dear Reviewer, we removed Figure 1 from the article since we considered that it no longer holds any relevance.
  2. b) Dear Reviewer, we added one new section in Table 1 that contains the indication of PGT per each study included if mentioned based on the study design. Moreover, we mentioned the number of participants allocated per groups analyzed to make the object of comparison.
  3. c) We think Table 1 would be more concise per data added into the manuscript.

Table 1. Stratification of studies based on the number of participants/couples, indications for PGT, and the reproductive outcomes.

Number of patients/

couples

Indications for PGT

Clinical outcomes

Reference

108 patients

n = 83; low-level mosaic SET

n = 25; high-level mosaic SET

recurrent IVF failure – (39.7%)

AMA – (49.4%)

RM – (13.2%)

male factor infertility – (4.8%)

combined preimplantation diagnosis – (7.2%)

unexplained infertility – (7.2%)

51.8% (n = 43) vs 52% (n = 13) – implantation

47% (n = 39) vs 52% (n = 13) – clinical pregnancy

47% (n = 39) vs 36% (n = 9) – ongoing pregnancy

5.1% (n = 2) vs 30.7% (n = 4) – miscarriage

44.6% (n = 37) vs 36% (n = 9) – live birth

[11]

108 patients

n = 42

n = 66

iRIF, RPL – (38.9%)

unexplained or male factor infertility – (61.1%)

76.2% (n = 48) vs 62.5% (n = 15) vs 25.0% (n = 5) – clinical pregnancy

79.4% (n = 50) vs 66.7% (n = 16) vs 25.0% (n = 5) – implantation

68.3% (n = 43) vs 62.5% (n = 15) vs 10.0% (n = 2) – ongoing pregnancy

[12]

155 patients

n = 17; <35yo

n = 14; < 35yo

n = 23; 35-37 yo

n = 23; 38-39yo

n = 56; 40-42yo

n = 25; ≥43yo

AMA – 58 (n = 37)

unexplained infertility – 33 (n = 21)

male factor infertility – 29 (n = 19)

ovulatory dysfunction – 11 (n = 7)

EMS – 6 (n = 4)

early RPL – 4 (n = 3)

tubal factor – 2 (n = 1)

other – 12 (n = 8)

80% (n = 68) – implantation rate

66% (n = 56) – clinical pregnancy rate

[13]

165 patients

36.5yo (3.5)

36.4yo (3.6)

aneuploidy – 160 (77) vs 139 (76)

AMA – 19 (9) vs 17 (9)

PGT-M/SR – 9 (4) vs 7 (4)

sex determination – 7 (3) vs 7 (4)

RPL – 6 (3) vs 5 (3)

desired SET – 5 (2) vs 5 (3)

recurrent IVF failure – 1 (0.5) vs 1 (0.5)

HLA determination – 1 (0.5) vs 1 (0.5)

67% (n = 101) – sustained implantation per embryo

61% (n = 79) – live birth per embryo

[14]

166 patients

n = 56; MTS-PGT-A

n = 90; MTS OD

n = 20; TS OD

No indication for PGT

22.58% vs 36.67% vs 16.67% – implantation per embryo transfer

41.9% (n = 13) vs 56.6% (n = 73) vs 29.2% (n = 7) – pregnancy rate per embryo transfer

20.0% (n = 13) vs 52.2% (n = 73) vs 28.0% (n = 7) – pregnancy rate per cycle

29.0% (n = 9) vs 45.7% (n = 64) vs 24.0% (n = 6) – clinical pregnancy

29.0% (n = 9) vs 40.7% (n = 57) vs 16% (n = 4) – ongoing pregnancy

43.3% vs 77.6% vs 29.2% – cumulative pregnancy

0% (n = 0) vs 0% (n = 0) vs 0% (n = 0) – ectopic pregnancy

30.9% (n = 4) vs 21.9% (n = 16) vs 42.8% (n = 3) – miscarriage

[15]

206 patients

n = 52; D3 zona opening

n = 63; D5 and D6 zona opening

AMA – 5 (9.62%) vs 12 (19.05%)

RIF – 7 (13.46%) vs 10 (15.87%)

RPL – 29 (55.77%) vs 25 (39.68%)

other – 11 (21.15%) vs 16 (25.40%)

64.71% (n = 22) vs 65.71% (n = 23) – clinical pregnancy

n = 1 vs n = 3 – miscarriage

[16]

228 patients

n = 81; non-ERA

n = 147; ERA

No indication for PGT

55.6% (n = 45) vs 56.5% (n = 83) – live birth

65.4% (n = 53) vs 67.4% (n = 99) – clinical pregnancy

14.8% (n = 9) vs 15.4% (n = 18) – biochemical pregnancy

13.2% (n = 7) vs 15.2% (n = 15) – miscarriage

[17]

255 patients

n = 153 Reciprocal translocation

(n = 77 ♀, n = 76 ♂)

n = 68 Robertsonian translocation

(n = 32 ♀, n = 36 ♂)

n = 25 Inversion

(n = 8 ♀, n = 17 ♂)

n = 4 Insertion

(n = 3 ♀, n = 1 ♂)

n = 1 Additional chromosome

(n = 0 ♀, n = 1 ♂)

n = 4 Multiple indications

(n = 3 ♀, n = 4 ♂)

repetitive implantation failure – 12.50% (n = 21)

RPL – 64.88% (n = 109)

AMA – 38.10% (n = 64)

SMF infertility – 3.57% (n = 6)

numerical abnormalities of the sex chromosome – 11.90% (n = 20)

previous trisomic pregnancy – 21.43% (n = 36)

68.04% (n = 132) – cumulative clinical pregnancy

59.79% (n = 116) – cumulative live birth/ongoing pregnancy

[18]

260 patients

n = 183; D5 blastocysts

n = 77; D6 blastocysts

No indication for PGT

75.4% (n = 138) vs 70.1% (n = 54) – pregnancy

67.8% (n = 124) vs 63.6% (n = 49) – implantation

57.9% (n = 106) vs 58.4% (n = 45) – ongoing pregnancy

[19]

270 patients

No allocation per group

No indication for PGT

63.10% (n = 53) vs 46.67% (n = 30) – ongoing pregnancy

[20]

274 patients

iRPL

n = 30; ≤35yo

n = 32; >35yo

Control

n = 157; ≤35yo

n = 55; >35yo

No indication for PGT

50.6% (n = 45) vs 63.7% (n = 128) – clinical pregnancy per transfer

50.0% (n = 46) vs 63.7% (n = 128) – implantation per transfer

4.5% (n = 4) vs 6.0% (n = 12) – biochemical pregnancy per transfer

24.4% (n = 11) vs 7.0% (n = 9) – clinical miscarriage per pregnancy

[21]

275 patients

n = 83; IVF-PGT-SR

n = 192; IVF-PGT-A

No indication for PGT

29.7% vs 15% – live birth

chromosome translocations patients

31.5% vs 28.8% – implantation

42.6% vs 25% – clinical pregnancy per PGT cycle

n = 17 vs n = 14 – live birth

aneuploidy patients

39.60% vs 29.6% – clinical pregnancy per PGT cycle

16.40% vs 20.50% – implantation

n = 66 vs n = 9 – live birth

[22]

283 patients

n = 100

n = 183

DOR – 47 (25.7%) vs 24 (24.0%)

EMS – 20 (10.9%) vs 5 (5.0%)

male factor infertility – 75 (41.0%) vs 25 (25.0%)

ovulatory dysfunction – 16 (8.7%) vs 18 (18.0%)

RPL – 0 (0.0%) vs 29 (29.0%)

tubal factor – 9 (4.9%) vs 12 (12.0%)

sex selection – 8 (4.4%) vs 5 (5.0%)

single gene disorder – 13 (7.1%) vs 3 (3.0%)

uterine factor infertility – 10 (5.5%) vs 6 (6.0%)

unexplained infertility – 19 (10.4%) vs 9 (9.0%)

other 11 (6.0%) vs 1 (1.0%)

52.3% (n = 148) – ongoing pregnancy

24.7% (n = 49) - pregnancy loss/ 7.6% (n = 15) – clinical and 17.2% (n = 34) – biochemical

[23]

296 patients

No allocation per group

AMA – 187 (63.2%)

AMA and RM – 36 (12.2%)

RM – 28 (9.5%)

RIF – 20 (6.7%)

AMA and RIF – 15 (5%)

embryo quality (excellent, good, average and poor)

85.7% (n = 12) vs 84.0% (n = 84) vs 80.0% (n = 44) vs 80.0% (n = 8) – implantation

7.1% (n = 1) vs 6% (n = 6) vs 5.4% (n = 3) vs 10% (n = 1) – clinical miscarriage

64.2% (n = 9) vs 66.0% (n = 66) vs 72.7% (n = 40) vs 60.0% (n = 6) – live birth

[24]

314 patients

No allocation per group

No indication for PGT

66.2% (n = 235) – pregnancy

52.4% (n = 186) – clinical pregnancy

50.1% (n = 178) – live birth

7.6% (n = 27) – biochemical pregnancy loss

0.6% (n = 2) – ectopic pregnancy

5.6% (n = 20) – early miscarriage

2.3% (n = 8) – late miscarriage

33.8% (n = 120) – no pregnancy

[25]

330 patients

No allocation per group

No indication for PGT

60% (n = 215) vs 40% (n = 8) – clinical pregnancy

53.8% (n = 192) vs 30% (n = 6) – live birth

18% (n = 65) vs 40% (n = 8) – miscarriage

[26]

364 patients

n = 98; AS

n = 266; non-AS

No indication for PGT

frozen/thawed embryo transfer with or without aneuploidy screening in the first embryo transfer attempt

64.29% (n = 63) vs 50.38% (n = 134) – implantation

3.17% (n = 2) vs 11.94% (n = 16) – miscarriage

6.12% (n = 6) vs 11.26% (n = 17) – biochemical pregnancy

53.06% (n = 52) vs 36.09% (n = 96) – live birth per patient

61.22% (n = 60) vs 43.98% (n = 117) – ongoing pregnancy/live birth per patient

frozen/thawed embryo transfer with or without aneuploidy screening in all transfer attempts

64.12% (n = 84) vs 51.60% (n = 226) – implantation

4.76% (n = 4) vs 12.39% (n = 28) – miscarriage

8.70% (n = 8) vs 9.96% (n = 25) – biochemical pregnancy

59.54% (n = 78) vs 41.78% (n = 183) – ongoing pregnancy/live birth per transfer

62.24% (n = 61) vs 50.38% (n = 134) – cumulative live birth per patient

79.59% (n = 78) vs 68.80% (n = 183) – cumulative ongoing pregnancy/live birth per patient

[27]

460 patients

No allocation per group

No indication for PGT

69.7% (n = 159) vs 57.2 (n = 63) – clinical pregnancy

[28]

525 patients

n = 325; vaginal delivery

n = 200; C-delivery

No indication for PGT

68.0% (n = 221) vs 55.5% (n = 111) – implantation

59.1% (n = 192) vs 49.0% (n = 98) – ongoing pregnancy and live birth

16.9% (n = 45) vs 20.1% (n = 28) – biochemical pregnancy

13.1% (n = 29) vs 11.7% (n = 13) – clinical miscarriage

[29]

554 patients

n = 497; PGT-A

n = 57; PGT-M

AMA

63.9% (n = 186) – clinical pregnancy

19.3% (n = 36) – miscarriage

50.5% (n = 147) – live birth

69.7% (n = 166) vs 38.5% (n = 10), 41.7% (n = 10), 0.0% (n = 0) – clinical pregnancy

18.7% (n = 31) vs 40.0% (n = 4), 10.0% (n = 1) – miscarriage

55.9% (n = 133) vs 23.1% (n = 6), 33.3% (n = 8), 0.0% (n = 0) – live birth

[30]

661 patients

n = 330; PGT-A group

n = 331; Control group

DOR – 8 (2.4%) vs 5 (1.5%)

ovulatory dysfunction – 77 (23.3%) vs 80 (24.2%)

tubal factor – 29 (8.8%) vs 29 (8.8%)

EMS – 17 (5.2%) vs 17 (5.1%)

uterine factor infertility – 7 (2.1%) vs 11 (3.3%)

other female factor infertility – 68 (20.6%) vs 72 (21.8%)

combined factor – 19 (5.8%) vs 21 (6.3%)

male factor infertility – 117 (35.5%) vs 121 (36.6%)

10.6% (n = 29) vs 8.3% (n = 26) – biochemical pregnancy

9.9% (n = 27) vs 9.6% (n = 30) – miscarriage

50.0% (n = 137) vs 45.7% (n = 143) – ongoing pregnancy

[31]

783 patients

No allocation per group

AMA – 576 (73.6%)

RIF – 32 (4.1%)

RPL – 28 (3.6%)

AMA + RIF – 27 (3.4%)

AMA + RPL – 13 (1.7%)

no indication – 107 (13.7%)

55.8% (n = 270) vs 55.0% (n = 155) vs 55.7% (n = 73) – positive pregnancy test

10.7% (n = 29) vs 12.3% (n = 19) vs 13.7% (n = 10) – biochemical pregnancy loss

12.0% (n = 29) vs 11.0% (n = 15) vs 12.7% (n = 8) – miscarriage

43.4% (n = 210) vs 42.9% (n = 121) vs 42.0% (n = 55) – live birth

[32]

974 patients

n = 256; <35yo

n = 220; 35-57yo

n = 227; 38-40yo

n = 118; 41-42yo

n = 65; >42yo

n = 108; donor

DOR – 23.8% vs 30.0% vs 36.1% vs 48.3% vs 53.8% vs 55.6%

male factor infertility – 28.5% vs 30.0% vs 22.5% vs 15.3% vs 7.7% vs 13.9%

PCOS – 10.9% vs 7.7% vs 4.8% vs 1.7% vs 4.6% vs 0.0%

uterine factor infertility – 2.3% vs 3.6% vs 5.7% vs 6.8% vs 9.2% vs 4.6%

tubal factor – 5.5% vs 2.7% vs 2.2% vs 5.1% vs 3.1% vs 0.9%

EMS – 1.2% vs 0.9% vs 1.8% vs 1.7% vs 0.0% vs 3.7%

other infertility condition – 12.9% vs 10.9% vs 14.1% vs 11.9% vs 10.8% vss 7.4%

idiopathic – 9.4% vs 6.4% vs 6.6% vs 4.2% vs 3.1% vs 0.0%

other – 5.5% vs 7.7% vs 6.2% 5.0% vs 1.5% vs 13.0%

69.9% vs 64.9% – implantation

n = 472 vs n = 94 – clinical pregnancies

49.6% vs 80.3% – clinical pregnancy per retrieval

70.6% vs 64.8% – clinical pregnancy per embryo transfer

n = 21 vs n = 6 – miscarriage

45.1% vs 65.0% – live birth per retrieval

64.5% vs 52.4% – live birth per embryo transfer

[33]

1051 patients

n = 589

n = 100

n = 362

male factor infertility – 18 (18.0%) vs 133 (22.6%)

PCOS – 13 (13.0%) vs 44 (7.5%)

EMS – 6 (6.0%) vs 37 (6.3%)

tubal factor – 6 (6.0%) vs 19 (3.2%)

unexplained infertility – 31 (31.0%) vs 170 (28.9%)

combined factor – 13 (13.0%) vs 106 (18%)

DOR – 13 (13.0%) vs 80 (13.6%)

RIF – 25 (25.0%) vs 212 (36.0%)

RM – 15 (15.0%) vs 47 (8.0%)

AMA – 11 (11.0%) vs 69 (11.7%)

combined factor – 49 (49.0%) vs 261 (44.3%)

56% (n = 589) – live birth

14.5% (n = 100) – miscarriage

34.4% (n = 362) – no clinical pregnancy/ n = 68 – biochemical pregnancy, n = 11 – ectopic pregnancy, n = 283 – negative pregnancy test

[34]

1152 patients

n = 225; DOR

n = 927; non-DOR

No indication for PGT

34.7% vs 34.2% – not pregnant

4.2% vs 3.9% – biochemical pregnancy

3.2% vs 6.8% – clinical miscarriage

1.1% vs 0.4% – ectopic pregnancy

56.8% vs 54.8% – live birth

[35]

1439 patients

n = 574; live birth

n = 561; no live birth

No indication for PGT

odds ratio 0.99; 95% confidence interval, 0.95-1.03 – implantation

odds ratio 0.98; 95% confidence interval, 0.94-1.01 – clinical pregnancy

odds ratio 1.03; 95% confidence interval, 0.95-1.12 – early pregnancy loss

odds ratio 0.99; 95% confidence interval, 0.95-1.03 – live birth

[36]

1513 patients

n = 744; SNP array

n = 769 - NGS

primary infertility – 171 (23.0%) vs 157 (20.4%)

secondary infertility – 564 (75.8%) vs 599 (77.9%)

tubal factor – 216 (29.0%) vs 302 (39.3%)

EMS – 56 (7.5%) vs 86 (11.2%)

ovulatory dysfunction – 33 (4.4%) vs 47 (6.1%)

oligospermia – 119 (16.0%) vs 127 (16.5%)

severe oligospermia – 39 (5.2%) vs 41 (5.3%)

OA – 5 (0.7%) vs 5 (0.7%)

51.34% (n = 306) vs 49.56% (n = 227) – implantation

9.56% (n = 57) vs 10.48% (n = 48) – biochemical pregnancy

10.07% (n = 60) vs 6.33% (n = 29) – spontaneous abortion

42.28% (n = 252) vs 44.1% (n = 202) – ongoing pregnancy/live birth

[37]

1531 patients

No allocation per group

AMA

repetitive implantation failure

RM

44.4% (n = 8) – clinical pregnancy

38.9% (n = 7) – ongoing pregnancy

65.9% vs 44.0%/ 57.4% vs 39.3% – clinical pregnancy and ongoing pregnancy in euploid compared to simple mosaic group

[38]

1629 patients

n = 1160

n = 469

No indication for PGT

54.0% (n = 491) vs 56.2% (n = 341) vs 43.2% (n = 311) – ongoing pregnancy/live birth – expansion 4 grade

55.6% (n = 857) vs 43.1% (n = 245) vs 32.3% (n = 41) – ongoing pregnancy/live birth – inner mass cell grade A

55.7% (n = 427) vs 52.8% (n = 531) vs 40% (n = 185) – ongoing pregnancy/live birth – trophectoderm grade

59.3% (n = 539) vs 61.8% (n = 375) vs 47.8% (n = 344) – clinical pregnancy – expansion 4 grade

60.5% (n = 933) vs 49.5% (n = 281) vs 34.6% (n = 44) – clinical pregnancy – inner mass cell grade A

60.1% (n = 461) vs 58.5% (n = 589) vs 44.9% (n = 208) – clinical pregnancy – trophectoderm grade

15.7% (n = 107) vs 17.1% (n = 84) vs 20.1% (n = 95) – early pregnancy loss – expansion 4 grade

14.3% (n = 169) vs 22.1% (n = 84) vs 39.8% (n = 33) – early pregnancy loss – inner mass cell grade A

16.5% (n = 97) vs 16.4% (n = 125) vs 21.8% (n = 64) – early pregnancy loss – trophectoderm grade

12.2% (n = 83) vs 13.4% (n = 66) vs 14.0% (n = 66) – clinical pregnancy loss – expansion 4 grade

13.1% (n = 155) vs 13.4% (n = 51) vs 10.8% (n = 9) – clinical pregnancy loss – inner mass cell grade A

10.9% (n = 64) vs 14.0% (n = 107) vs 15.0% (n = 44) – clinical pregnancy loss – trophectoderm grade

[39]

1884 patients

n = 646; Control - ≤60d

n = 599; 61-90d

n = 679; 91-180d

n = 405; 181-360d

n = 144; 361-720d

n = 118; 721-1080d

n = 97; >1080d

No indication for PGT

42.9% (n = 291), 41.7% (n = 169), 39.6% (n = 57) vs 49.4% (n = 319) – live birth

[40]

1997 patients

n = 846; STEET after IVF/PGT-A with aCGH

n = 1151; STEET after IVF/PGT-A with NGS

No indication for PGT

1.30% (n = 11) vs 0.7% (n = 8) – clinical error per embryo transfer

2.04% (n = 11) vs 1.01% (n = 8) – clinical error per pregnancy

0.43% (n = 2) vs 0.14% (n = 1) – clinical error per ongoing pregnancy/live birth

23.33% (n = 7) vs 13.33% (n = 6) – clinical error per spontaneous abortion with products of concenption available

[41]

3480 patients

n = 155; <18.5 BMI

n = 2549; 18.5-24.9 BMI

n = 591; 25-29.9 BMI

n = 185; ≥30

AMA vs. RPL; 1.20

RIF vs. RPL; 1.16

MF vs. RPL; 1.21

Other vs. RPL; 1.08

Cycle type, modified natural vs. HRT; 0.94

22.7% – miscarriage

implantation, pregnancy, clinical pregnancy and live birth rates calculated but not properly specified

[42]

31649 patients

No allocation per group

No indication for PGT

62% – clinical pregnancy

57% – combined mean ongoing pregnancy (n = 813) and live birth (n = 1057) per transfer

9% – biochemical, spontaneous miscarriage and later term loss

[43]

100 couples

No allocation per group

No indication for PGT

69.23% (n = 36) – biochemical pregnancy

65.38% (n = 34) – clinical pregnancy

2.94% (n = 1) – miscarriage

n = 8 – ongoing pregnancy

[44]

142 couples

n = 142; (89.31%) blastocysts had low mtDNA copy number

n = 17; blastocysts had hugh mtDNA copy number

No indication for PGT

59.15% (n = 84) vs 47.10% (n = 8) – pregnancy

13.38% (n = 19) vs 17.65% (n = 3) – biochemical miscarriage

45.77% (n = 65) vs 29.41% (n = 5) – implantation

6.15% (n = 4) vs 40.0% (n = 2) – miscarriage

42.96% (n = 61) vs 17.65% (n = 3) – ongoing pregnancy

[45]

150 couples

n = 56; PAI carriers

n = 94; PEI carriers

No indication for PGT

67.4% (n = 29) vs 67.5% (n = 54) – clinical pregnancy per embryo transfer number

3.4% (n = 1) vs 14.7% (n = 8) – spontaneous abortion

6.9% (n = 2) vs 1.8% (n = 1) – ectopic pregnancy

[46]

180 couples

n = 124; Reciprocal translocation

n = 21; Inversion

n = 35; Robertsonian translocation

No indication for PGT

12.24% (n = 12) vs 18.18% (n = 4) vs 11.36% (n = 5) – biochemical pregnancy per embryo transfer cycle

54.08% (n = 53) vs 59.09% (n = 13) vs 47.73% (n = 21) – clinical pregnancy per embryo transfer cycle

13.21% (n = 7) vs 15.38% (n = 2) vs 19.05% (n = 4) – miscarriage per clinical pregnancy

0% (n = 0) vs 7.69% (n = 1) vs 0% (n = 0) – ectopic pregnancy per clinical pregnancy

n = 9 vs n = 2 vs n = 2 – ongoing pregnancy

[47]

206 couples

n = 102; PGT-A group

n = 104; Control group

NOA – 16 (15.7%) vs 26 (25.0%)

OA – 5 (4.9%) vs 5 (4.8%)

OAT-S – 81 (79.4%) vs 73 (70.2%)

tubal factor – 16 (15.7%) vs 21 (20.2%)

ovulation dysfunction – 30 (29.4%) vs 32 (30.8%)

EMS – 5 (4.9%) vs 6 (5.8%)

66.7% (n = 60) vs 69.8% (n = 74) – clinical pregnancy per transfer

6.7% (n = 4) vs 21.6% (n = 16) – early miscarriage per transfer

62.2% (n = 56) vs 54.7% (n = 58) – ongoing pregnancy per transfer

54.9% (n = 56) vs 55.8% (n = 58) – cumulative ongoing pregnancy per transfer

[48]

265 couples

n = 184; Group A

n = 81; Group B

tubal factor – 48 vs 40

male factor infertility – 14 vs 17

anovulation – 23 vs 17

other (EMS, unexplained infertility, combined factors) – 15 vs 26

64.7% (n = 66) vs 43.5% (n = 10) – pregnancy per transfer

48.0% (n = 49) vs 39.1% (n = 9) – clinical pregnancy per transfer

51.0% (n = 52) vs 39.1% (n = 9) – implantation per transfer

7.8% (n = 8) vs 4.3% (n = 1) – miscarriage per transfer

[49]

648 couples

n = 402; study group

n = 1208; Control group

DOR – 5.98% vs 10.93%

male factor infertility – 23.51% vs 24.01%

tubal factor – 8.76% vs 7.86%

ovulatory dysfunction – 20.32% vs 19.78%

unexplained infertility – 38.25% vs 29.97%

64.7% (n = 202) vs 0% (n = 0) vs 68.8% (n = 11) vs 30.8% (n = 12) – implantation

1.0% (n = 3) vs 0% (n = 0) vs 0% (n = 0) vs 0% (n = 0) – ectopic pregnancy

7.4% (n = 23) vs 23.5% (n = 24) vs 12.5% (n = 2) vs 5.1% (n = 2) – clinical miscarriage

9% (n = 28) vs 16.7% (n = 17) vs 12.5% (n = 2) vs 12.8% (n = 5) – biochemical loss

17.9% (n = 56) vs 59.8% (n = 61) vs 6.3% (n = 1) vs 51.3% (n = 20) – not pregnant

73.1% vs 23.5% 82.1% – clinical pregnancy

[50]

779 couples

n = 633; <40 yo

n = 240; normal

n = 264; overweight

n = 129; obese

n = 146; ≥40yo

n = 48; normal

n = 57; overweight

n = 41; obese

No indication for PGT

<40y

67.9% (n = 106) vs 69.6% (n = 117) vs 75.6% (n = 68) – implantation

51.2% (n = 66) vs 47.4% (n = 65) vs 62.2% (n = 51) – pregnancy loss

40.4% (n = 63) vs 42.9% (n = 72) vs 34.4% (n = 31) – live birth

≥40y

86.8% (n = 33) vs 78.4% (n = 29) vs 46.4% (n = 13) – implantation

44.7% (n = 17) vs 64.7% (n = 22) vs 69.2% (n = 18) – pregnancy loss

55.3% (n = 21) vs 32.4% (n = 12) vs 28.6% (n = 8) – live birth

[51]

1418 couples

NGS vs SNP array

n = 157 vs 105 – Robertsonian translocation

n = 405 vs 342 – Reciprocal translocation

n = 40 vs 14 – Inversion

n = 111 vs 94 – PGS

n = 92 vs 58 – Sex chromosome abnormality

No indication for PGT

50.5% (n = 341) vs 41.7% (n = 228) – clinical pregnancy

15.5% (n = 54) vs 22.8% (n = 52) – miscarriage

[52]

SET – single embryo transfer, AMA – advanced maternal age, RM – recurrent miscarriage, iRIF – idiopathic repeated implantation failure, RPL – recurrent pregnancy loss, EMS – endometriosis, HLA – human leukocyte antigen, MTS – mosaic Turner syndrome, OD – oocyte donation, SMF – severe male factor, ERA – endometrial receptivity assay DOR – diminished ovarian reserve, AS – aneuploidy screening, PCOS – polycystic ovarian syndrome, OA – obstructive azoospermia, STEET - single thawed euploid embryo transfer, BMI – body mass index, HRT – hormonal replacement therapy, PAI – paracentric, PER – pericentric, NOA – nonobstructive azoospermia, OAT-S – severe oligoasthenoteratozoospermia

Comments from the Reviewer:

  1. Discussion
  2. It’s a little bit short for the discussion. The significance of this review for clinicians should be emphasized.

Response:

Dear Reviewer, the following paragraph has been added into the manuscript:

“Conclusively, numerous factors may impact reproductive outcomes in a double-sword manner, which is why clear inclusion criteria are compulsory to obtain optimal results. Even though several predictors are implied in such protocols, errors still might occur but the sensitivity and specificity of these platforms could be amplified. Even though national programs should become a priority regardless of the institution, several interventions should be avoided due to associated risks. Fortunately, NGS is a groundbreaking tool in a plethora of research fields, through which new evidence has been brought in a number of pathologies. This is why NGS must be implemented in all laboratories that conduct ARTs. From our point of view, NGS now stands as the main barrier toward a new stage in our understanding of genetic defects.”

Kind regards and all the best,

Ovidiu-Dumitru Ilie